NLRP3 inflammasome and pyroptosis: implications in inflammation and multisystem disorders

Li Xiaodi 1
Zhang Zhiyuan 1
Han Yang 2 13820093454@163.com
Zhang Mianzhi 2 2895346749@qq.com
1 Tianjin University of Traditional Chinese Medicine , Tianjin , China
2 Tianjin Academy of Traditional Chinese Medicine Affiliated Hospital , Tianjin , China
Uversky Vladimir
Electronic publication date: 2025 Aug 15
Publication date: 2025
Volume: 13
Electronic Location ID: e19887
Received 2025 Jan 29; Accepted 2025 Jul 20
Copyright: © 2025 Li et al.
Copyright year: 2025
Copyright holder: Li et al.
License: This is an open access article distributed under the terms of the Creative Commons Attribution License, which permits unrestricted use, distribution, reproduction and adaptation in any medium and for any purpose provided that it is properly attributed. For attribution, the original author(s), title, publication source (PeerJ) and either DOI or URL of the article must be cited.
License URL: https://creativecommons.org/licenses/by/4.0/

Keywords: Pyroptosis, NLRP3 inflammasome, NEK7, Palmitoylation of GSDMD-NT, Ninjurin-1, Innate immunity, Multisystem disease

Funding: Tianjin Municipal Health Commission Chinese Medicine and Western Medicine Research Project 2023205 Zhang Mianzhi Tianjin Famous Traditional Chinese Medicine Inheritance Studio New Kidney Kangning Capsule Joint Research Project This work was supported by the Tianjin Municipal Health Commission Chinese Medicine and Western Medicine Research Project (grant number: 2023205), Zhang Mianzhi Tianjin Famous Traditional Chinese Medicine Inheritance Studio, and New Kidney Kangning Capsule Joint Research Project. The funders had no role in study design, data collection and analysis, decision to publish, or preparation of the manuscript.

==============================
The NLRP3 inflammasome is a multiprotein complex that senses diverse pathogen-associated molecular patterns (PAMPs) and danger-associated molecular patterns (DAMPs), activating the pyroptosis pathway. Pyroptosis is a form of programmed cell death that plays a crucial role in immune responses and inflammatory processes. The NLRP3 inflammasome-gasdermin D (GSDMD) axis has emerged as a critical therapeutic target in inflammatory diseases. Oligomerization of NLRP3 triggers caspase-1 activation, which subsequently induces GSDMD palmitoylation—an essential event that facilitates pyroptosis. Clinically, NLRP3 inhibitors, such as MCC950, demonstrate protective effects in NLRP3-mediated inflammatory diseases. GSDMD holds substantial potential as a diagnostic, monitoring, and therapeutic target across diverse diseases, underscoring its utility as a pan-biomarker. This review aims to synthesize current knowledge regarding the structure and function of the NLRP3 inflammasome and the regulatory mechanisms governing pyroptosis. Additionally, integrating findings from multiple physiological systems highlights the key roles of the NLRP3 inflammasome and pyroptosis in disease pathogenesis, offering novel perspectives for targeting inflammatory responses and associated disorders.

Introduction

In 2002, the Tschopp laboratory first proposed the concept of the inflammasome, which is a multi-protein complex composed of NLRP3 protein, apoptosis-associated speck-like protein (ASC), and pro-cysteine-aspartic protease-1 (pro-caspase-1) (Martinon, Burns & Tschopp, 2002). D’Souza & Heitman (2001) first proposed the term “pyroptosis” in their study of Salmonella infection in macrophages to define a specific type of programmed cell death associated with inflammatory responses. Activation of the NLRP3 inflammasome cleaves GSDMD, liberating its N-terminal domain (NT). Following palmitoylation modification, the NT binds to membrane lipids and oligomerizes to form pores in the plasma membrane. The pores formed by the GSDMD and Ninjurin-1 (NINJ-1) protein play a synergistic role in pyroptosis, inducing changes in intracellular osmotic pressure. During the execution phase of pyroptosis, cells undergo gradual swelling, culminating in plasma membrane rupture (PMR). The release of the cell’s contents not only disrupts the integrity of the cell but also activates the immune system, triggering the process of pyroptosis (Shi et al., 2015; Ding et al., 2016).

Pyroptosis is a distinct form of cell death with unique physiological and pathological roles compared to apoptosis and necrosis. Beyond its fundamental significance in biological research, pyroptosis holds substantial translational promise for clinical disease diagnosis and therapeutic development. Elucidating pyroptotic mechanisms, regulatory networks, and disease involvement is crucial for advancing preventive strategies, therapeutic interventions, and novel drug discovery. Consequently, this review is primarily intended for researchers and clinicians specializing in cell biology, medicine, and biotechnology.

Having established the mechanistic basis of NLRP3 inflammasome activation and GSDMD-executed pyroptosis in the Introduction, it is essential to define the methodological framework for synthesizing evidence on their multisystem implications. The following section details our systematic literature survey strategy, ensuring comprehensive, objective, and reproducible analysis. This foundation supports our subsequent examination of molecular mechanisms (NLRP3 assembly to pyroptosis execution) and their dysregulation across physiological systems in disease contexts.

Survey methodology

This comprehensive review systematically examines the roles of the NLRP3 inflammasome and pyroptosis in multisystem diseases, we conducted systematic searches across major academic databases, including Google Scholar, PubMed, and Web of Science, to ensure broad literature coverage. Key search terms comprised: “NLRP3 inflammasome,” “gasdermin D,” “pyroptosis,” and “multisystem disorders.” We systematically searched for studies focusing on the roles of the NLRP3 inflammasome and pyroptosis in these diseases. During the screening process, we applied specific exclusion criteria to ensure the relevance and quality of the selected content. The search was optimized through: (1) removal of duplicate articles; (2) full-text evaluation against predefined criteria: relevance (direct mechanistic or therapeutic focus on NLRP3 inflammasome/pyroptosis in multi-system diseases), methodological rigor (study design appropriateness, e.g., controlled experiments, adequate sample size); (3) prioritization of high-impact studies. Only the most relevant and up-to-date literature was included, except landmark papers. We excluded studies that did not clearly state therapeutic effects or did not focus on the role of pyroptosis in the pathogenesis of the diseases. After a comprehensive evaluation, we identified 120 articles that met the inclusion criteria, prioritizing the most recent and impactful studies, which formed a strong foundation for our review.

Upstream of pyroptosis: The NLRP3 inflammasome

Pattern recognition receptors constitute crucial components for detecting inflammatory signals, specifically enabling recognition of PAMPs and DAMPs, thereby triggering innate immune responses (Kanneganti, 2020). Figure 1 illustrates the process whereby the NLRP3 protein, upon perceiving these specific molecular patterns, activates the NLRP3 inflammasome and promotes its assembly, a critical step for initiating pyroptosis.

Figure 1 Upstream signaling pathway leading to pyroptosis activation.

(1) Upon detection of PAMPs or DAMPs, the NLRP3 inflammasome complex assembles and oligomerizes. (2) This oligomerized NLRP3 inflammasome recruits and activates pro-caspase-1. (3) Activated caspase-1 cleaves pro-IL-1β and pro-IL-18 into their mature, bioactive forms (IL-1β, IL-18). (4) Activated caspase-1 simultaneously cleaves GSDMD, releasing its GSDMD-NT. (5) GSDMD-NT oligomerizes and inserts into the plasma membrane, forming pores that lead to cell swelling (pyroptotic morphology), release of inflammatory cytokines (IL-1β, IL-18), and ultimately, lytic cell death (pyroptosis). The graphics were created using BioGDP.com.

Composition of the NLRP3 inflammasome

NLRP3 is a member of the nucleotide-binding domain leucine-rich repeat containing protein family (Fu & Wu, 2023). The proteins of this family mainly participate in regulating immune responses in the human body, especially inflammation responses (Chou et al., 2023). ASC, as a soluble protein, primarily localizes to the nucleus of cells in the absence of external stimuli. Upon sensing infection or inflammatory signals, ASC redistributes from the nucleus to the cytoplasm (Liu et al., 2022a). Therein, the protein undergoes self-oligomerization to form speckles and engages in binding with NLRP3. Subsequently, NLRP3 recruits pro-caspase-1 and orchestrates the assembly of the NLRP3 inflammasome. During this process, pro-caspase-1 undergoes proteolytic cleavage to generate two distinct subunits: p20 and p10. These subunits subsequently associate to form a p20/p10 heterodimer, which exhibits partial catalytic activity. Fully active caspase-1 is achieved through the assembly of multiple heterodimers into a tetrameric complex (Luksch et al., 2015).

The fully activated caspase-1 can cleave two specific protein precursors, namely pro-IL-1β and pro-IL-18, and convert them into biologically active forms-though not necessarily simultaneously (Ghiringhelli et al., 2009; Zaki et al., 2010). Following receptor binding, IL-1β and IL-18 activate the NF-κB signaling pathway (Lalor et al., 2011). As a key transcriptional regulator, NF-κB controls the expression of genes central to immune and inflammatory responses (Xiao & Ghosh, 2005).

Essential regulator of the NLRP3 inflammasome activation and assembly: NIMA-related kinase 7

The activation of the NLRP3 inflammasome involves multiple factors, yet its precise mechanism remains poorly characterized. Current investigations focus on unraveling the cellular signaling networks and molecular determinants that modulate the interaction between NLRP3 and NIMA-related kinase 7 (NEK7).

The NIMA family exerts diverse functions in cell biology, including regulating the growth of cilia, maintaining microtubule stability, facilitating mitosis, supporting cell growth, and participating in DNA damage responses (Bachus et al., 2022). As a member of this kinase family, NEK7 exerts core regulatory functions in inflammasome activation (Zhao et al., 2020). NLRP3-activating signals induce intracellular potassium efflux, which facilitates NLRP3-NEK7 interaction and ultimately drives both assembly and functional activation of the NLRP3 inflammasome (He et al., 2016). Potassium efflux through ion channels coupled with mitochondrial dysfunction coordinately triggers mitochondrial reactive oxygen species (ROS) production and chloride intracellular channel (CLIC)-dependent translocation to the cytosol. Subsequently, NEK7 activates NLRP3 through the cooperative action of potassium efflux, ROS signaling, and CLIC activity (Liu et al., 2020).NLRP3 activation enables its leucine-rich repeat (LRR) domain to engage NEK7, relieving autoinhibition of the nucleotide-binding and oligomerization domain (NACHT) (Shi et al., 2016). This promotes NACHT oligomerization, thereby driving inflammasome assembly. Post-activation phosphorylation of NEK7, dependent on potassium efflux and GSDMD pore formation, enhances NLRP3-NEK7 interaction to establish a positive feedback loop that specifically drives inflammasome activation cascades (Xu et al., 2025). Concurrent studies demonstrate that phosphorylation dynamics at Ser803 of NLRP3 regulate NEK7 recruitment to the inflammasome complex (Niu et al., 2021).

Previous studies have shown that the DOC domain of the HECT family E3 ubiquitin ligase HECTD3 interacts with the NACHT/LRR domain of NLRP3, thereby inhibiting the activation of the NLRP3 inflammasome. This finding unveils a new regulatory mechanism for the activation of the NLRP3 inflammasome and proposes HECTD3 as a novel therapeutic target for related diseases (Cheng et al., 2024).

Downstream of pyroptosis: GSDMD-mediated membrane pores

The Gasdermin family contains members such as Gasdermin A, B, C, D, E, and DFNB59. Among these, DFNB59 is the only member lacking a pore-forming active domain, whereas others typically exhibit this function (Shi et al., 2015).

After being cleaved by caspase, GSDMD executes a critical downstream event in pyroptosis by translocating to and perforating the cell membrane through the formation of gasdermin pores. This key pore-forming activity, which disrupts membrane integrity and initiates the lytic cell death characteristic of pyroptosis, is schematically depicted in Fig. 2. GSDMD consists of an NT and a C-terminal autoinhibitory domain (CT). The CT domain constitutively suppresses NT-mediated pore formation through intramolecular interactions (Liu et al., 2016). During pyroptosis, GSDMD-NT specifically binds to lipids in the cell membrane, creating transmembrane pores that allow the release of intracellular contents into the extracellular space (Schaefer & Hummer, 2022). Lipid interactions are a key step in pore formation (Hodel et al., 2021). However, the cleavage of GSDMD alone is not sufficient to form pores (Du et al., 2023). Therefore, further research is needed to investigate other factors that may act synergistically to trigger PMR.

Figure 2 Molecular mechanisms of pyroptotic pore formation and effector release.

Following pyroptosis initiation, GSDMD-NT undergoes palmitoylation and oligomerization catalyzed by ZDHHC5/9 enzymes, forming transmembrane pores in the plasma membrane. These pores facilitate the release of pro-inflammatory cytokines (IL-1β, IL-18) and ions (K+, Ca2+). Concomitantly, GSDMD pore formation triggers NINJ1 activation, which oligomerizes to form secondary pores enabling the efflux of larger DAMPs such as LDH and HMGB1. Collectively, GSDMD-NT and NINJ1 coordinate the lytic release of inflammatory mediators, executing the cell death program. The graphics were created using BioGDP.com.

Palmitoylation of GSDMD-NT

Pyroptosis is primarily mediated by two pathways: the canonical pathway and the non-canonical pathway. Both pathways depend on GSDMD as a common downstream effector (Wei, Feng & Zhang, 2022). GSDMD is widely expressed in diverse tissues (Broz, Pelegrín & Shao, 2020). Previous studies have shown that palmitoylation of GSDMD-NT plays a crucial role in triggering pyroptosis (Zhang et al., 2024c). For example, inhibiting palmitoylation of human GSDMD at the C191 site using the specific inhibitor NU 6300 effectively suppressed GSDMD cleavage in the human monocytic leukemia cell lines (Jiang et al., 2024b). This further indicates that palmitoylation serves as a critical regulatory mechanism for human GSDMD activation and pyroptosis execution. Palmitoylation is a protein modification process that modulates immune regulatory networks by altering the subcellular localization, lifespan, and intermolecular interactions of pyroptosis-related proteins (Jiang et al., 2024a). ZDHHC5 and ZDHHC9 are essential enzymes for GSDMD palmitoylation (Stine & Humphries, 2024). When the NLRP3 inflammasome is activated, GSDMD undergoes palmitoylation at a specific site (Cys191 in humans and Cys192 in mice) in a ROS-dependent manner (Foley, 2024). Palmitoylation is a key step for GSDMD to facilitate effective pore formation during pyroptosis (Sun & Hornung, 2024). Inhibition of GSDMD palmitoylation significantly attenuates macrophage pyroptosis and IL-1β release, thereby ameliorating organ injury and enhancing survival in septic mice (Balasubramanian et al., 2024). This finding further confirms that palmitoylation of GSDMD-NT plays an important role in regulating membrane localization and activation of GSDMD.

Pyroptosis-related PMR executor: NINJ1

NINJ1 is a recently identified PMR protein that amplifies inflammation by fragmenting pyroptotic cells into smaller debris, which are subsequently phagocytosed and cleared by other host cells (Pandey et al., 2021). It acts via its extracellular alpha helix structure penetrates the plasma membrane and embeds into it, promoting the aggregation of NINJ1 monomers to form amphiphilic filamentous structures, which ultimately leads to PMR (Broz, 2023). NINJ1 is involved in multiple forms of cell death, including pyroptosis, necrosis, and apoptosis (Kayagaki et al., 2021; Imre, 2024). However, its role varies across different cell death modalities (Ramos et al., 2024).

Kayagaki et al. (2021) found that following cell death stimulation, NINJ1 aggregates and oligomerizes on the cell surface, which is a key event in PMR and occurs downstream of GSDMD pore formation (Dias, Hornung & Nylandsted, 2022). This event poses a challenge to the passive process of PMR during cell death. The trigger for NINJ1 oligomerization in pyroptosis remains unclear (Newton et al., 2024). Some researchers propose that NINJ1 oligomerization may be related to membrane permeabilization, membrane lipid redistribution, or membrane tension (Kayagaki et al., 2021). Some researchers also believe that it may be related to ionic influx (David et al., 2024). NINJ1 and GSDMD are structurally distinct pore-forming proteins that differ fundamentally in their size exclusion properties. NINJ1 generates size-unrestricted pores, mediating the release of large molecules such as lactate dehydrogenase (LDH) and high mobility group box 1 (HMGB1) (Volchuk et al., 2020; Kayagaki et al., 2021). Conversely, GSDMD forms size-restricted pores that permit exclusively small molecules to pass (Degen et al., 2023). Through coordinated action, these proteins collectively execute pyroptosis. Although NINJ1 is critical for PMR, it is not essential for GSDMD pore formation (Kayagaki et al., 2021). As a key regulator, NINJ1 effectively controls and facilitates the release of inflammatory cytokines by mediating PMR, thereby fine-tuning the pyroptosis-driven inflammatory response and shaping the organism’s immune homeostasis.

Following the detailed description of NLRP3 inflammasome activation (upstream triggers/assembly) and GSDMD-mediated pyroptosis execution (pore formation), the following section synthesizes current research on the dysregulation of this pathway and its critical roles in inflammation and multisystem disorders pathogenesis.

Research progress on the NLRP3 inflammasome and pyroptosis in multiple systems

The NLRP3 inflammasome is intimately linked to the pathogenesis of multisystem diseases, particularly by driving pyroptosis, which exacerbates inflammatory responses. Research on the NLRP3 inflammasome and pyroptosis has advanced considerably in recent years, particularly within neurological, endocrine, and urinary systems. These advancements and the consequent identification of promising therapeutic targets are summarized and visually represented in Fig. 3, providing a consolidated view for developing novel treatment approaches.

Figure 3 Current research advances on the role of pyroptosis in multiple systemic diseases.

Recent findings on pyroptosis across major organ systems, including neurological, endocrine, and urinary disorders, detailing its involvement in disease pathomechanisms, the identification of novel therapeutic targets, and the development of associated pharmacological agents. It underscores the pivotal contribution of pyroptosis to multi-organ dysregulation and its significant potential as a druggable pathway for future interventions. The graphics were created using BioGDP.com.

Nervous system

The activation of the NLRP3 inflammasome and its driving of pyroptosis are intimately linked to the pathogenesis of Alzheimer’s disease (AD) (Rui et al., 2021). Therefore, targeting this mechanism may offer novel therapeutic avenues for AD. Indeed, this hypothesis is supported by studies on Rhodiola glycosides in AD therapy, which exerted its therapeutic effects by inhibiting NLRP3 inflammasome-mediated pyroptosis (Cai et al., 2021). Pyroptosis plays a pivotal role in the pathogenesis of Parkinson’s disease (PD). Targeted inhibition of the NLRP3 inflammasome attenuates neuroinflammation and reduces the loss of dopaminergic neurons. Consequently, NLRP3 inhibitors represent a promising therapeutic approach for PD (Sadier et al., 2025). Furthermore, α-lipoic acid protects dopaminergic neurons effectively by suppressing NLRP3 inflammasome activation and its mediated GSDMD pathway, concurrently downregulating S100A9 protein expression, thereby emerging as a novel therapeutic strategy for PD (Saad, Atef & Elsayed, 2025; Zhang et al., 2025). In metabolic disorders, high cholesterol levels can trigger inflammasome activation, leading to neuronal pyroptosis (de Dios et al., 2023). This finding unveils a potential mechanistic link between metabolic dysregulation and neurodegenerative diseases. Additionally, intermittent theta burst stimulation (iTBS) has been shown to attenuate neuronal pyroptosis and improve motor function in cerebral ischemia mice via the TLR4/NFκB/NLRP3 signaling axis (Luo et al., 2022).

Targeting pyroptosis with nanoparticle technology can effectively alleviate neuroinflammation and prevent secondary injury after traumatic brain injury, which is of significant importance in the field of neuroinflammation and injury (Zhang et al., 2024b). It is worth noting that aluminum exposure has been shown to induce central nervous system damage by activating the NLRP3-mediated pyroptosis pathway (Hao et al., 2023). Therefore, interventions targeting this mechanism may offer potential strategies for mitigating the adverse effects of aluminum exposure.

Studies on epilepsy and stroke have shown that the neuronal nitric oxide synthase (nNOS)/ROS pathway is involved in pyroptosis in epilepsy, and inhibiting this pathway may confer neuroprotective benefits on epilepsy patients (Xu et al., 2023). Concurrently, studies have revealed that targeting the pyroptosis pathway could emerge as a novel therapeutic strategy for the prevention and treatment of stroke (Long et al., 2023).

Endocrine system

Previous studies have demonstrated an intimate association between islet β-cells and pyroptosis within the endocrine system. In diabetic patients, pyroptosis promotes diabetes progression and accelerates pancreatic β-cell failure (Roshanravan et al., 2020). Increased GSDMD expression in the pancreatic tissue of diabetic mice provides evidence for the critical role of pyroptosis in diabetic pathogenesis (Li et al., 2022a). Intestinal-derived exosomes enriched with angiotensin-converting enzyme 2 restore pancreatic β-cell function by suppressing NLRP3 inflammasome activation and β-cell pyroptosis (Yang et al., 2024). Moreover, exercise training ameliorates insulin resistance and hepatic injury in elderly prediabetic patients by inhibiting NLRP3 inflammasome activation (Zhang et al., 2023).

In thyroid diseases, cadmium exposure induces pyroptosis in thyroid follicular cells, compromising thyroid tissue integrity and endocrine function (Chen et al., 2023a). Studies demonstrate that Ubiquitin-specific peptidase 1 (USP1) drives Hashimoto’s thyroiditis (HT) progression through deubiquitination-mediated stabilization of NLRP3 and promotion of its transcription, which significantly enhances inflammasome-mediated pyroptosis. Consequently, USP1 inhibition effectively blocks the pyroptotic pathway and attenuates HT-associated pathological damage (Zhao et al., 2024). Research on polycystic ovary syndrome (PCOS) has yielded new insights. Specifically, the NLRP3 inflammasome-dependent pathway not only drives pyroptosis but also modulates glycan biosynthesis, sex hormone biosynthesis, autophagy, and apoptosis in granulosa cells (Wang et al., 2024). Mogroside V ameliorates PCOS by suppressing pyroptosis and insulin resistance, thereby restoring granulosa cell function and follicular development (Yang et al., 2025). Exposure to di-(2-ethylhexyl) phthalate induces granulosa cell pyroptosis by activating the SLC39A5/NF-κB/NLRP3 signaling pathway, thereby impairing ovarian function (Sun et al., 2023). In addition, histone deacetylase 1 inhibits granulosa cell pyroptosis by regulating the H19/miR-29a-3p/NLRP3 axis through its deacetylase activity (Chen et al., 2023b). These studies reveal the mechanistic link between granulosa cell pyroptosis and ovarian dysfunction, whereby distinct signaling pathways govern cellular fate decisions, thereby determining ovarian pathophysiology.

Urinary system

In the research field of urinary system diseases, pyroptosis not only induces inflammatory injury but also acts as a potential mediator in the progression of acute kidney injury to chronic kidney disease (Tajima et al., 2019; Jiang et al., 2021; Li et al., 2022b). In particular, in diabetic kidney disease (DKD) research, studies have shown that elevated neutrophil extracellular traps promote dysfunction and pyroptosis of glomerular endothelial cells, thereby accelerating disease progression (Zheng et al., 2022). In addition, studies have demonstrated that the (pro)renin receptor induces pyroptosis of DKD renal tubular epithelial cells via the dipeptidyl peptidase 4-mediated signaling pathway (Xie et al., 2024). It is worth noting that novel lncRNA-prader willi/Angelman region RNA, SNRPN neighbor aggravates tubular epithelial pyroptosis by regulating thioredoxin-interacting protein through dual ways in DKD (Song et al., 2023). Conversely, deficiency of stimulator of interferon genes suppresses NLRP3 inflammasome activation, thereby mitigating podocyte inflammatory responses and exerting protective effects against pyroptosis in DKD (Yang et al., 2023). Additionally, accumulation of dihydroxyacetone phosphate is also considered another important factor that may lead to podocyte pyroptosis in DKD (Zhang et al., 2024a).

Recent studies have demonstrated that circular RNA and pyroptosis have played important roles in the pathogenesis of DKD. For example, circ_0004951 promotes pyroptosis in DKD renal tubular epithelial cells through the miR-93-5p/NLRP3 inflammasome axis (Wang et al., 2022a). circACTR2 is pivotal for diabetic tubular cell survival by regulating high glucose-induced pyroptosis in proximal tubular cells (Wen et al., 2020). MiR-4449 is essential for DKD pathogenesis, modulating IL-1β/IL-18 expression, ROS levels, and pyroptosis (Gao et al., 2021).

Research indicates that a high-salt diet may serve as a significant potential risk factor for overactive bladder (OAB) (Xue et al., 2024), potentially by impairing the integrity of the bladder epithelial barrier and activating the NLRP3 inflammasome signaling pathway, thereby inducing an inflammatory response. This cascade can subsequently lead to detrusor overactivity. These mechanisms suggest that restoring bladder epithelial barrier function and inhibiting NLRP3-associated inflammatory responses may represent potential therapeutic targets for OAB. Furthermore, hispidulin can ameliorate cyclophosphamide-induced cystitis by suppressing NLRP3 inflammasome activation, achieved through targeting prostaglandin G/H synthase 2 (PTGS2) (Liu et al., 2024b).

These findings have underscored the complex and crucial pyroptotic mechanisms in urinary system diseases, particularly in DKD. They provide a theoretical foundation for the development of novel therapeutic approaches, offering promise to improve the prognosis and quality of life of DKD patients.

Digestive system

The NLRP3 inflammasome and pyroptosis are essential in digestive system diseases. These findings provide new scientific foundations and potential therapeutic targets for the management of inflammatory disorders and tumors of the digestive system.

Research on inflammatory disorders of the digestive system: the NLRP3 inhibitor Selnoplast has shown good safety and efficacy in patients with ulcerative colitis, offering novel therapeutic prospects for inflammatory bowel disease (Klughammer et al., 2023). The expression changes of Discs Large MAGUK Scaffold Protein 2 in intestinal inflammation and colorectal cancer regulate inflammasome assembly, which correlates with disease progression (Keane et al., 2022). By suppressing pro-caspase-1 recruitment, anthocyanins inhibit NLRP3 inflammasome activation in non-alcoholic fatty liver disease, providing a theoretical basis for anti-inflammatory therapeutic development (Zhu et al., 2022). Furthermore, these phytochemicals attenuate inflammatory progression through integrated mechanisms involving decreased NF-κB activity, reduced IL-1 and IL-6 secretion, and promoted anti-inflammatory pathways (Afsar et al., 2022). Atractylenolide I (ATR-I) exerted its therapeutic effect against gastric ulcers (GU) by significantly mitigating local inflammatory responses in an indomethacin-induced rat gastric ulcer model. This was achieved through pronounced inhibition of the NLRP3 inflammasome signaling pathway, as evidenced by downregulation of mRNA and protein expression levels of NLRP3, ASC, caspase-1, and NF-κB (Yuan et al., 2024). Similarly targeting the NLRP3 pathway, vitexin effectively alleviated 1-methyl-3-nitro-1-nitrosoguanidine-induced chronic atrophic gastritis. Vitexin directly inhibited the NLRP3 inflammasome activation, leading to a significant reduction in pro-inflammatory cytokine production and amelioration of gastric histopathological damage (Liu et al., 2025). Collectively, these studies demonstrate that targeted inhibition of the NLRP3 inflammasome represents a promising therapeutic strategy for the gastric inflammatory disorders investigated in these models, namely gastric ulcer and chronic atrophic gastritis.

Research on tumors of the digestive system: nitrosamines induce pyroptosis of the human esophageal epithelial cells via the NLRP3/caspase-1/GSDMD axis, unveiling the carcinogenic potential of these chemicals (Liu et al., 2022b). Notably, NLRP3 inflammasome activation drives proliferation and migration of esophageal squamous cell carcinoma, offering new therapeutic perspectives for esophageal cancer (Yu et al., 2020).

Respiratory system

The NLRP3 inflammasome represents a key focus in respiratory disease research. As NLRP3 inflammasome inhibitors, DFV 890 and Tranilast have demonstrated promise in pneumonia therapy by alleviating virus-induced inflammation and enhancing antiviral efficacy (Gatlik et al., 2024; Saeedi-Boroujeni et al., 2022; Madurka et al., 2023). The NLRP3 inflammasome represents a promising target for combination therapy, with potential applications in both COVID-19 management and intervention for fertility issues (Bazrafkan et al., 2021). COVID-19 is an infectious condition caused by severe acute respiratory syndrome coronavirus 2 (SARS-CoV-2). Studies have demonstrated that the SARS-CoV-2 nucleocapsid protein can activate the complement system via the mannose-binding lectin-dependent pathway, with subsequent activation of this signaling cascade shown to induce NLRP3 inflammasome activation (Zhao, Di & Xu, 2021). Furthermore, experimental evidence indicates that SARS-CoV2-derived double-stranded RNA and single-stranded RNA can be specifically recognized by endosomal Toll-like receptor 3, Toll-like receptor 7, and melanoma differentiation-associated gene 5. This molecular recognition triggers activation of the NF-κB signaling pathway, thereby upregulating gene expression of NLRP3 inflammasome-associated components along with precursor proteins of IL-1β and IL-18 (Yin et al., 2023). The hyperactivation of the NLRP3 inflammasome within the reproductive system has been implicated in the pathogenesis of various reproductive disorders, including infertility, miscarriage, pre-eclampsia, and PCOS, through its induction of inflammatory mediator imbalance characterized by dysregulated pro-inflammatory and anti-inflammatory responses (Hosseini et al., 2023). In mechanical ventilation research, low tidal volume ventilation attenuates lung injury by suppressing NLRP3 inflammasome activation (Wang et al., 2022c). In rat models of acute lung injury, studies have shown that pyroptosis is involved in the pathophysiology (Wang et al., 2022b). In lung fibrosis research, studies have shown that crystalline silica accelerates lung fibrosis progression by inducing pyroptosis and modulating autophagy (Wei et al., 2023). Suppressing GSDMD-dependent pyroptosis could mitigate lung fibrosis progression (Song et al., 2022).

Research indicates that the NLRP3 inflammasome is critical in driving the pathogenesis of asthma. Specifically, NLRP3 levels in induced sputum are significantly elevated in children with asthma, with further elevation observed in those with moderate-to-severe disease, suggesting that induced sputum NLRP3 may serve as a potential biomarker for assessing asthma severity (Li & Liu, 2024). Notably, environmental factors such as PM2.5 exposure can upregulate the ROS/NF-κB signaling pathway in airway epithelial cells, thereby activating the NLRP3 inflammasome, which mediates pyroptosis and ultimately leads to airway tissue damage (Kang et al., 2024). Collectively, these findings underscore the importance of the NLRP3 inflammasome in asthma pathophysiology and environmentally induced asthma pathogenesis. Studies have also revealed that NLRP3 inflammasome–mediated epithelial pyroptosis contributes to fibrotic healing following tracheal injury (Li et al., 2023). Additionally, in research on Aspergillus fumigatus infection, studies have shown that irradiated mice exhibit enhanced susceptibility to Aspergillus fumigatus via the NLRP3/GSDMD pathway in bronchial epithelium (Wu et al., 2022). These findings not only enhance our understanding of the NLRP3 inflammasome’s role in respiratory diseases but also furnish novel insights for future treatment strategies.

Cardiovascular system

In cardiovascular disease research, the NLRP3 inflammasome and associated pathways are central drivers of vascular pathologies, particularly in myocardial injury and cardiomyopathy (Liu et al., 2024a). For example, GSDMD-mediated pyroptosis of cardiomyocytes drives myocardial ischemia/reperfusion injury (MI/RI) (Shi et al., 2021). The E3 ubiquitin ligase MARCH 2 suppresses the PGAM5/MAVS/NLRP3 axis to inhibit pyroptosis, thereby protecting the myocardium from MI/RI (Liu et al., 2024a). In diabetic cardiomyopathy studies, methyltransferase-like protein 14 has been shown to decelerate cardiomyopathy progression by suppressing pyroptosis (Meng et al., 2022).

Moreover, mouse models have demonstrated that mitochondrial damage and activation of the cyclic GMP-AMP synthase (cGAS)-stimulator of interferon genes pathway induce cardiomyocyte pyroptosis and myocardial hypertrophy (Yan et al., 2022).

Previous studies have identified CircRNA DICAR as a novel endogenous regulator involved in diabetic cardiomyopathy and cardiomyocyte pyroptosis in diabetes (Yuan et al., 2023). Studies have shown that activation of the NLRP3 inflammasome induces cardiomyocyte pyroptosis via caspase-1, thereby impairing myocardial function and contributing to dilated cardiomyopathy (Zeng et al., 2020). Regarding myocardial infarction, mechanically induced pyroptosis improves the resistance and metabolic capacity of cardiosphere-derived cells to oxidative stress, holding therapeutic potential for myocardial infarction (Wang et al., 2023). Recent studies on septic cardiomyopathy have revealed that targeted modulation of NLRP3 inflammasome activation represents an effective therapeutic strategy. Specifically, Vaccarin alleviates sepsis-induced myocardial injury by enhancing NLRP3 palmitoylation, thereby inducing its conformational inactivation (Zhu et al., 2024). Another significant approach involves cortistatin, which activates the SSTR2-AMPK-Drp1 signaling axis to suppress mitochondrial fission and reduce ROS levels, ultimately blocking the NLRP3 inflammasome-dependent cardiomyocyte pyroptosis pathway and conferring protection against septic cardiomyopathy (Duan et al., 2024). Collectively, these studies underscore the pivotal role of the NLRP3 pathway in septic cardiomyopathy pathogenesis and provide novel directions for targeted interventions.

Novel inhibitors targeting NLRP3 and GSDMD pathways

The NLRP3 inflammasome and GSDMD pathways play pivotal roles in the regulation of inflammatory responses and pyroptosis, making them attractive targets for therapeutic intervention in a wide range of diseases, including autoimmune disorders, neurodegenerative diseases, and cardiovascular diseases. Recent advancements in pharmacological research have led to the development of novel inhibitors that specifically target these pathways, offering new hope for treating these complex conditions. In this section, we will explore the latest findings on inhibitors targeting NLRP3 and GSDMD, starting with the well-studied MCC950 as a prime example of NLRP3-targeting agents, followed by an analysis of key inhibitory strategies against GSDMD.

Targeting NLRP3: the case of MCC950

MCC950 (CP-456,773) is a sulfonylurea derivative that selectively inhibits IL-1β biosynthesis (Perregaux et al., 2001). It has garnered attention as a potent inhibitor of the NLRP3 inflammasome (Mekni et al., 2019). MCC950 precisely targets the NLRP3 inflammasome without exerting significant effects on other related inflammatory signaling pathways (Coll et al., 2015). Mechanistically, MCC950 binds to a specific cleft in the NACHT domain of NLRP3, thereby blocking inflammasome activation and suppressing inflammatory responses (Hochheiser et al., 2022). It is a promising candidate for inhibiting NLRP3-driven pathologies. Substantial progress in MCC950 research has highlighted its therapeutic potential for inflammation-associated diseases (Zheng et al., 2024). To realize this potential, developing high-efficacy/low-toxicity derivatives and novel inhibitors is essential to overcome limitations and synergistically improve target selectivity, safety, efficacy, and druggability.

Beyond these developments, additional NLRP3 inhibitors have emerged. The discovery of NP3-253, a pyridazinone-phenol chemotype, provides structurally novel NLRP3 inhibitors with distinct mechanisms of action. This breakthrough has enabled the development of brain-penetrant candidate molecules, offering new therapeutic avenues for treating neuroinflammatory diseases (Harrison, 2024). GDC-2394, an NLRP3 inflammasome inhibitor, has undergone a first-in-human Phase I study in healthy volunteers to evaluate its safety, pharmacokinetics, and pharmacodynamics (Tang et al., 2023). Similarly, ZYIL1-an orally bioavailable NLRP3 inflammasome inhibitor-has progressed through Phase I clinical trials (Parmar et al., 2023).

Targeting GSDMD: key inhibitory strategies

Recent advances in GSDMD-targeted therapies highlight novel therapeutic strategies. The small-molecule inhibitor GI-Y1 specifically blocks pyroptotic pore formation, thereby attenuating cardiomyocyte pyroptosis and mitigating ischemia-reperfusion injury (Zhong et al., 2023). The next-generation candidate GI-Y2 significantly suppresses atherosclerosis progression through inhibition of the pyroptosis pathway (Fan et al., 2025). Collectively, these agents provide innovative approaches for cardiovascular disease prevention and treatment. Furthermore, NU6300 impedes GSDMD activation via a dual mechanism: covalent binding to Cys191 and suppression of palmitoylation. This action obstructs membrane translocation and oligomerization of GSDMD, presenting new directions for treating inflammatory disorders (Jiang et al., 2024b).

Conclusions

This review systematically consolidates recent mechanistic advances in NLRP3 inflammasome activation and the execution of pyroptosis. Critically, studies across diverse disease models have underscored the pivotal role of this pathway in inflammatory pathologies affecting the nervous, endocrine, and urinary systems, highlighting the immense therapeutic potential of targeting the NLRP3 inflammasome or its downstream effectors. However, significant knowledge gaps persist, including: the spatiotemporal dynamics of pathway activation; the pathophysiological significance of key post-translational modifications and their mechanisms in dynamically regulating pathway activity; tissue/cell-type-specific differences in pathway regulation; challenges in targeted delivery and safety for clinical translation. Future research must prioritize: leveraging advanced technologies to dissect mechanistic details, developing tissue-specific interventions, elucidating the pathway’s crosstalk with other cell death or inflammatory signaling pathways, and accelerating the clinical translation of mechanism-based therapeutics. Addressing these critical gaps is essential for realizing the promise of targeting the NLRP3/pyroptosis pathway to improve human health.

Supplemental Information

Supplemental Information 1 All references in this literature review.

Abbreviations

AD Alzheimer’s disease

Al aluminum

ASC apoptosis-associated speck-like protein

ATR-I atractylenolide I

CAG chronic atrophic gastritis

CARD the caspase recruitment domain

Cd cadmium

cGAS-STING the cyclic GMP-AMP synthase (cGAS)-stimulator of interferon genes

CLIC chloride intracellular channel

CST cortistatin

CT C-terminal

DAMPs danger-associated molecular patterns

DEHP di-(2-ethylhexyl) phthalate

DHAP dihydroxyacetone phosphate

DKD diabetic kidney disease

ESCC esophageal squamous cell carcinoma

GSDMD gasdermin D

GU gastric ulcers

HDAC1 histone deacetylase 1

Het-1A human esophageal epithelial cells

HMGB1 high mobility group box protein B1

HT Hashimoto’s thyroiditis

iTBS intermittent theta burst stimulation

LDH lactate dehydrogenase

LRR leucine-rich repeat

MI myocardial ischemia

NACHT the nucleotide-binding and oligomerization domain

NAFLD nonalcoholic fatty liver disease

NEK7 NIMA-related kinase 7

NETs neutrophil extracellular traps

NINJ-1 ninjurin-1

NT N-terminal, merged for duplicates

OAB overactive bladder

PAMPs pathogen-associated molecular patterns

PCOS polycystic ovary syndrome

PD Parkinson’s disease

PMR plasma membrane rupture

pro-caspase-1 pro-cysteine-aspartic protease-1

PRR the (Pro) retin receptor

PTGS2 prostaglandin G/H synthase 2

PWARSN Prader Willi/Angelman Region RNA, SNRPN Neighbor

RI reperfusion injury

ROS reactive oxygen species

SARS-CoV-2 severe acute respiratory syndrome coronavirus 2

STING stimulator of interferon genes

UC ulcerative colitis

USP1 ubiquitin-specific peptidase 1

Additional Information and Declarations

Competing Interests

The authors declare that they have no competing interests.

Author Contributions

Xiaodi Li conceived and designed the experiments, performed the experiments, analyzed the data, prepared figures and/or tables, authored or reviewed drafts of the article, and approved the final draft.

Zhiyuan Zhang analyzed the data, prepared figures and/or tables, and approved the final draft.

Yang Han conceived and designed the experiments, prepared figures and/or tables, authored or reviewed drafts of the article, and approved the final draft.

Mianzhi Zhang conceived and designed the experiments, prepared figures and/or tables, authored or reviewed drafts of the article, and approved the final draft.

Data Availability

The following information was supplied regarding data availability:

This literature review did not generate raw data.

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
