# Peer review of "NLRP3 inflammasome and pyroptosis: implications in inflammation and multisystem disorders"

_PeerJ, doi:10.7717/peerj.19887_

## Round 0.1 · original submission · Major Revisions

**Language Note:** The review process has identified that the English language must be improved. PeerJ can provide language editing services - please contact us at [email protected] for pricing (be sure to provide your manuscript number and title). Alternatively, you should make your own arrangements to improve the language quality and provide details in your response letter. – PeerJ Staff

Reviewer 1 ·

Basic reporting

.

Experimental design

.

Validity of the findings

.

Additional comments

This review on NLRP3 inflammasome and pyroptosis in multisystem diseases has major flaws and add little to the topic of NLRP3 inflammasome and pyroptosis. The following needs to be addressed:
1. Abstract is lengthy without specific translational information.
2. Figures need to be improved and descriptive legends have be added.
3. Overall, the text is narrative and superficial, lacking in depth analysis and has too many typos.
4. The translation of recent targets of NLRP3 is very limited for different inflammatory disease; Moreover, the first inhibitor, MC950 is not cited. Furthermore, there are several novel inhibitors of NLRP3, GSDMD; which are not shown.
5. List of abbreviations is necessary.

·

Basic reporting

Dear respected editor;
Regarding the article titled (The NLRP3 Inflammasome and Pyroptosis in Multisystem Diseases)
This research designed is good and the methodology well explained. Also, the results and conclusion are clear, but there are some comments need to be revised.
I am suggesting this title; "NLRP3 Inflammasome and Pyroptosis: Implications in Inflammation and Multisystem Disorders"
Major Comments:
1. Design & Structure
• The manuscript is well-structured and provides an in-depth analysis of the NLRP3 inflammasome and its role in multisystem diseases. However, some sections could benefit from improved organization for enhanced clarity and readability.
• The transition between sections is sometimes abrupt, particularly in the "Survey Methodology" and "Research Progress on the NLRP3 Inflammasome and Pyroptosis in Multiple Systems" sections. A clear transition statement summarizing previous points and introducing the new topic would improve coherence.
• The manuscript lacks clear subsection headings in the discussion of different organ systems (e.g., Nervous System, Endocrine System, etc.). Although each system is discussed separately, better formatting (bold headings or numbered subsections) would improve readability.
• Figures and tables are mentioned but not adequately referenced in the text (e.g., "Figure 1"). If they are included in the final version, they should be explicitly referred to in relevant paragraphs with explanations.
2. Language & Grammar
• The language is generally comprehensive but overly technical, sometimes making it difficult for a broader scientific audience to grasp key concepts quickly. Simplifying some complex sentences without losing depth could enhance clarity.
• Numerous grammatical errors, awkward phrasing, and typographical mistakes are present throughout the manuscript. Some examples include:
o Abstract:
 "Pyroptosis is a programmed cell death that plays a crucial role in immune response and inflammation processes."
 Suggested revision: "Pyroptosis is a form of programmed cell death that plays a crucial role in immune responses and inflammatory processes."
 "At the same time, combined with the research progress of multiple physiological systems, the key roles of the NLRP3 inflammasome and pyroptosis in the pathogenesis of diseases have been revealed..."
 Suggested revision: "Additionally, integrating findings from multiple physiological systems highlights the key roles of the NLRP3 inflammasome and pyroptosis in disease pathogenesis..."
o Introduction:
 "pyroptosis is a novel form of cell death that plays a unique role in physiological and pathological processes compared to traditional apoptosis and necrosis."
 "Pyroptosis" should be capitalized at the beginning of the sentence.
 More concise phrasing could improve readability:
"Pyroptosis is a distinct form of cell death with unique physiological and pathological roles compared to apoptosis and necrosis."
o Survey Methodology:
 "The search was optimized by removing duplicate articles, evaluating the full text for usability, and prioritizing high-impact studies."
 The phrase "evaluating the full text for usability" is vague. Consider specifying criteria for usability.
3. Scientific Rigor
• The manuscript is well-referenced and includes an extensive citation list, which supports claims and findings.
• However, some statements need stronger justification with direct references. For instance:
o In the "Upstream of Pyroptosis: The NLRP3 Inflammasome" section, the mechanism by which the NEK7 protein regulates NLRP3 activation is not fully detailed. Including more molecular details or referring to specific studies would improve credibility.
o The "Digestive System" section briefly mentions anthocyanins inhibiting NLRP3 activation in non-alcoholic fatty liver disease but lacks mechanistic details.
o The "Respiratory System" section states, "The NLRP3 inflammasome is a potential combination therapy target that can be used for both COVID-19 management and intervention for fertility issues." This is a broad claim and should be backed by specific studies explaining the mechanisms in both cases.
4. Conceptual & Logical Issues
• Some redundancy exists in the description of GSDMD, NINJ1, and NLRP3, leading to repetition of similar concepts in different sections.
• The "Conclusions" section repeats many findings instead of providing insights or future directions. Consider focusing on:
o The therapeutic implications of targeting the NLRP3 inflammasome.
o Knowledge gaps that require further research.
o How different disease models have contributed to current understanding.
o Please rewrite it profisionally.
5. References
• The reference list appears comprehensive but should be formatted according to the target journal’s citation style. Please manege it to be unified for the all references according to journal format.

Final Assessment & Suggestions for Improvement
1. Revise the manuscript for grammatical accuracy and technical clarity.
2. Improve transitions between sections for better logical flow.
3. Enhance formatting with clear subsection headings in the disease-specific discussions.
4. Ensure all claims are well-supported with direct references.
5. Reduce redundancy in descriptions of molecular mechanisms (especially for inflammasome activation and pyroptosis execution).
6. Strengthen the conclusion by focusing on future directions and therapeutic prospects.
This review identifies critical areas for improvement while acknowledging the manuscript’s scientific depth and relevance. With structural and linguistic refinements, it can achieve a higher standard of clarity, rigor, and impact in the field.

Experimental design

Good, need some improvement.

Validity of the findings

I think it is acceptable

Additional comments

No

---

## Round 0.2 · accepted · Accept

We have been unable to get the original reviewers to re-review this revision, but in my view, all the issues pointed out by the reviewers were adequately addressed, and the manuscript was revised accordingly. The revised version is acceptable now.